# Spike-timing-dependent plasticity learning of coincidence detection with passively integrated memristive circuits

M. Prezioso[1], M.R. Mahmoodi[1], F. Merrikh Bayat[1], H. Nili[1], H. Kim[1], A. Vincent[1] & D.B. Strukov[1]

Spiking neural networks, the most realistic artificial representation of biological nervous systems, are promising due to their inherent local training rules that enable low-overhead online learning, and energy-efficient information encoding. Their downside is more demanding functionality of the artificial synapses, notably including spike-timing-dependent plasticity, which makes their compact efficient hardware implementation challenging with conventional device technologies. Recent work showed that memristors are excellent candidates for artificial synapses, although reports of even simple neuromorphic systems are still very rare. In this study, we experimentally demonstrate coincidence detection using a spiking neural network, implemented with passively integrated metal-oxide memristive synapses connected to an analogue leaky-integrate-and-fire silicon neuron. By employing spike-timing-dependent plasticity learning, the network is able to robustly detect the coincidence by selectively increasing the synaptic efficacies corresponding to the synchronized inputs. Not surprisingly, our results indicate that device-to-device variation is the main challenge towards realization of more complex spiking networks.

---

[1] Electrical and Computer Engineering Department, UC Santa Barbara, Santa Barbara, CA 93106-9560, USA. These authors contributed equally: M. Prezioso, M. R. Mahmoodi, F. Merrikh Bayat. Correspondence and requests for materials should be addressed to D.B.S. (email: strukov@ece.ucsb.edu)

The development of spiking neural networks (SNNs) has been largely driven by the hope of replicating the extremely high energy efficiency of biological systems. The main hypothesis behind the energy efficiency is the SNN's considerably higher information capacity (e.g., in terms of Vapnik–Chervonenkis dimension), compared to the more popular firing-rate-type neural networks such as multi-layer perceptron[1–3]. Moreover, training for the firing-rate networks typically relies on a backpropagation algorithm, the efficient implementation of which is challenging owing in part to the centralized method for computing weight updates and the requirement for large high-precision memory. By contrast, most popular SNNs' weight updates rules are local, requiring only information from pre- and post-synaptic neurons (e.g., see Eq. 2 below), which could be a significant advantage for compact and low power implementations of real-time training, and scaling towards more complex networks[4].

In the simplest SNN models, the information is encoded in spike−time correlations[1,2], while the network functionality is defined by the neural link strengths, i.e., the synaptic efficacies, which are adjusted based on the relative timing of spikes that are passed via synapses. A key network element is a leaky-integrate-and-fire (LIF) neuron, whose operation is described by the equation:

$$C\frac{dU(t)}{dt} = \sum_i G_i V_i^{pre}(t) - \frac{1}{R_L}U(t), \qquad (1)$$

where $U$, $C$, and $R_L$ mimic the membrane potential, capacitance, and its leakage resistance, correspondingly. The dot product in Eq. (1) is a current flowing into the neuron, which is proportional to the sum of the individual synaptic currents, i.e., products of pre-synaptic spike voltages $V^{pre}(t)$ and corresponding synaptic conductances (weights) $G$. The second term on the right-hand side of Eq. (1) is a membrane leakage current. Note that $R_L C$ is membrane characteristic time, which determines the integration rate of the LIF neuron. Additionally, Eq. (1) does not explicitly include the resting potential. Instead, the resting potential is used as a reference for $U(t)$.

Another important SNN feature is spike-timing-dependent plasticity (STDP), which is a timing-dependent specialization of Hebbian learning[1,2,5,6]. A typical goal of STDP learning is to strengthen the synaptic efficiency when two events happen in the expected causal temporal order, and to weaken it otherwise. STDP rule formally describes the change in synaptic weight as a specific function $f_{STDP}$ of a difference in firing times between pre-synaptic ($t^{pre}$) and post-synaptic ($t^{post}$) spikes, i.e.,

$$\Delta G = f_{STDP}(t^{pre} - t^{post}). \qquad (2)$$

For example, a very common STDP rule found in biological synapses of neocortex layer 5 is shown in Fig. 1c[6], according to which weight is increased (decreased) when pre-synaptic spike reaches the synapse before (after) the post-synaptic one.

Compact, low power hardware implementation of STDP learning is of particular importance, because practically valuable neural networks require a massive number of synapses. Owing to the extremely high density and analogue functionality, resistive switching memory devices ('memristors') are considered as one of the most attractive candidates for artificial synapses[7]. Indeed, crossbar circuit arrays with passively integrated memristors[8,9] enable the most efficient hardware for dot-product computation[10]. Moreover, STDP learning in memristors can be implemented by applying specific bipolar pulses (see Fig. 1)[11]. There have been many experimental demonstrations of various STDP rules[12–18] and more advanced synaptic functionalities

based on memristors[19–21] (also see comprehensive reviews in refs. [22,23]). However, memristor implementations of even simple SNNs, which rely on STDP learning, are so far very scarce[24–28]. For example, Milo et al. used nine discrete memristors, connected in 1 transistor + 1 resistor (1T1R) configuration, to demonstrate learning of $3 \times 3$ patterns[24]. In ref. [25], Kim et al. reported SNN implementation based on CMOS-integrated 2T1R phase-change memory devices. Their demonstration did not include LIF neurons, so that the post-spikes were triggered externally after post-processing the neuron input. The focus of ref. [26] was on emulating homoeostasis using organic-based devices immersed in an electrolyte. However, the demonstrated coincidence detection, i.e. detection of the occurrence of temporally close but spatially distributed input signals, was performed without any learning. Ambrogio et al. reported the experimental STDP dynamics for hafnium-oxide memristors and used these results to simulate an SNN based on 1T1R discrete memristors[27]. A more advanced demonstration of STDP-based spatial pattern recognition was demonstrated more recently by the same group, though still based on the same discrete 1T1R memristors[28].

Here, we experimentally demonstrate operation and STDP learning in SNN implemented with the most prospective, passively integrated (0T1R) memristive synapses connected to a silicon LIF neuron. Similar to ref. [26], the focus of our work is on coincidence detection, i.e., the task of identifying correlated spiking activity. Coincidence detection functionality is found in various parts of nervous systems, such as auditory[29,30] and visual[31] cortices and is generally assumed to play a very important role in the brain[32–34]. In fact, entire neural systems have been hypothesized based purely on synchrony[35].

## Results

**Experimental setup**. The implemented SNN features 20 input neurons connected via 20 memristive crossbar-integrated synapses to a single LIF neuron (Fig. 2). Figure 3 shows particular voltage spike shapes employed in the coincidence detection experiment. (More details on the experimental setup and spike parameters are reported in Methods section.) The specific amplitudes and durations of voltage pulses were chosen to ensure balanced STDP windows, with approximately 50−200% maximum changes in conductance, for a majority of the devices. The post-synaptic spike has a larger amplitude negative pulse, which was needed owing to larger absolute values of reset switching thresholds compared to those of set transition.

In the coincidence detection task, STDP learning mechanism is employed for training a single neuron to fire an output spike when it receives simultaneous (i.e., correlated) spikes from several input neurons. Figure 2a shows an example of a spike pattern applied to the rows of the crossbars in the experiment. The correlated input spikes, i.e., rows 11 through 15 in Fig. 2a, were always synchronized and repeated every 840 ms, which represents one 'frame' of spiking input. Furthermore, in some experiments, the position of each synchronous spike was randomly shifted by $t_{jitter}$ to emulate temporal jitter. The uncorrelated ('noise') inputs were approximated by randomly generating spikes, with a probability $p_{spike}$ for each row in each frame. The position of these spikes within the frame was also generated probabilistically, by adding a random offset $\Delta_{offset}$ to the specific reference time, corresponding to the shifted spike phase—see Supplementary Figure 1 for more details.

**Coincidence detection results**. At the beginning of coincidence detection experiments, conductances of all memristors were set to approximately 16 μS (Figs. 4e–h, 5b) using an automatic tuning algorithm[36], which roughly corresponds to the middle of the device's dynamic range[8,9]. Figure 4 summarizes evolution of the

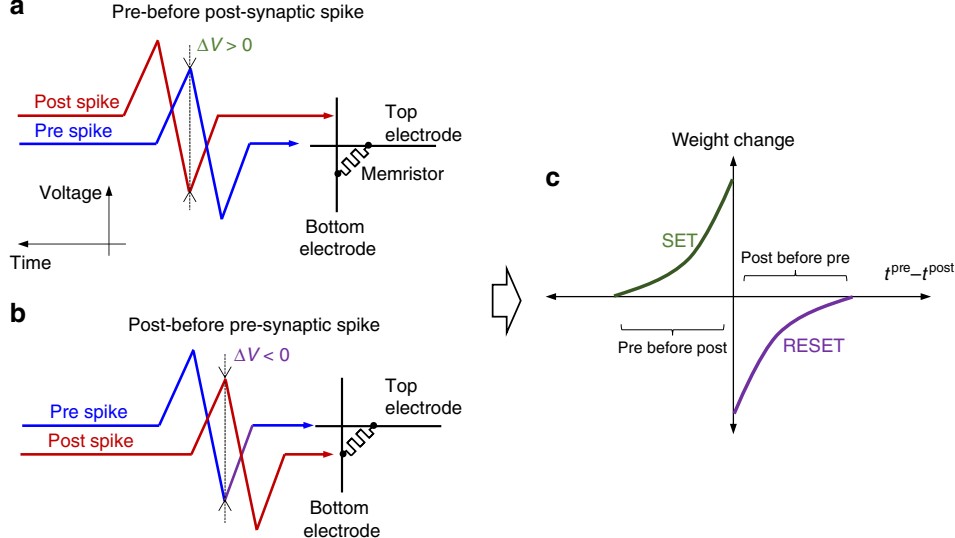

**Fig. 1** STDP implementation. Relative timing of the pre- and post-synaptic spikes for performing **a** potentiation and **b** depression of the synaptic weight for the memristor-based implementation[11], resulting in **c** the specific STDP, which is commonly observed in biology

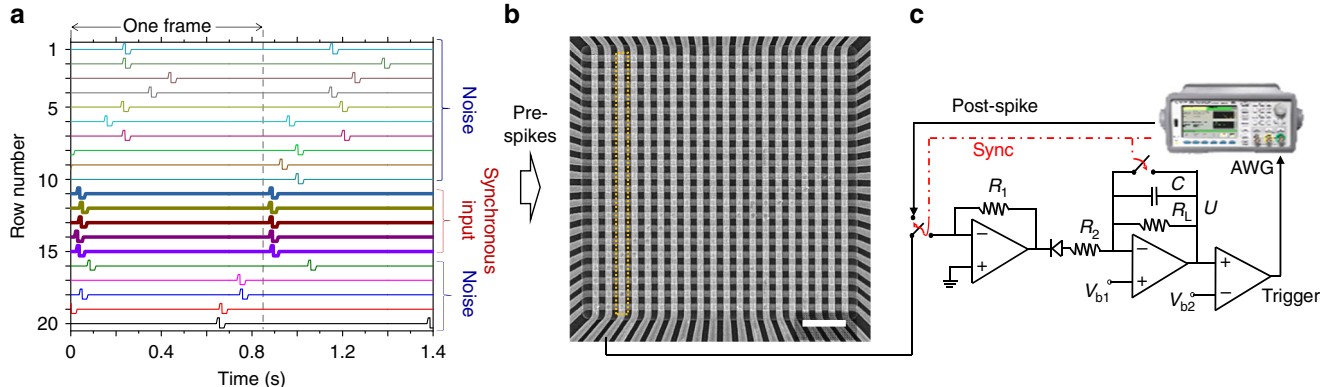

**Fig. 2** Spiking neural network setup. **a** Example of pattern of spikes applied to the inputs of the crossbar circuit. **b** Scanning electron top-view image of the utilized crossbar circuit. Scale bar is 2 μm. **c** LIF neuron implementation. See Methods section for more details on the crossbar circuit and neuron implementations

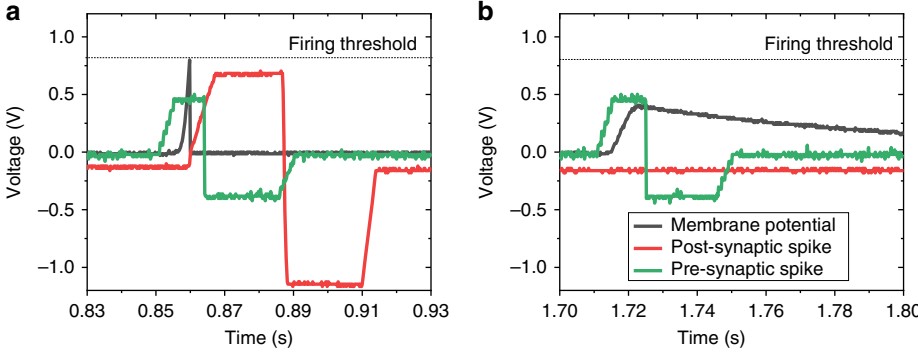

**Fig. 3** Signal waveforms. An example of measured oscilloscope signals in the lower-noise pattern coincidence experiment, showing **a** a pre-synaptic synchronous spike, that along with other synchronous inputs, led to the generation of the postsynaptic spike, and **b** an out-of-phase pre-synaptic synchronous spike with correspondingly increasing and then gradually leaking membrane potential

system for the first set of experiments. The left panels show specific frames of spiking input, while the right panels show the trajectory of each synaptic weight (during 20 epochs or a total of 1.68 s of spiking activity for Fig. 4e–g and 60 epochs for Fig. 4h), with thick/thin lines corresponding to the synchronized/ uncorrelated inputs. (One epoch is defined as an application of ten frames. In each frame the spikes have the same reference positions with different random jitter and offsets.) Specifically, for all these experiments, $p_{\text{spike}} = 1$ so that the average spiking activity for all inputs was 1.19 Hz. To investigate the impact of

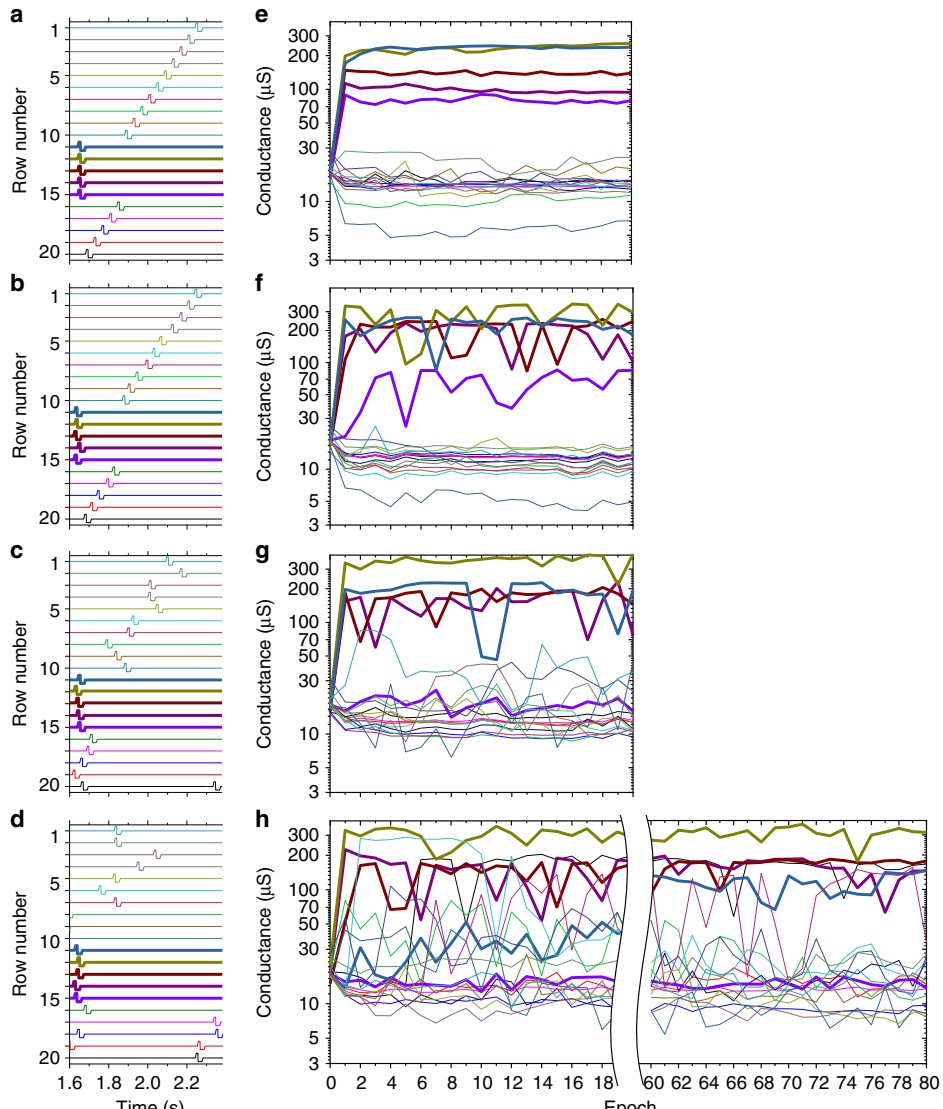

**Fig. 4** Coincidence detection of higher-noise pattern. **a–d** Example of specific spiking input frame with **a** $(t_{jitter})_{max} = 0$, $(\Delta_{offset})_{max} = 0$, **b** $(t_{jitter})_{max} = 27$ ms, $(\Delta_{offset})_{max} = 27$ ms, **c** $(t_{jitter})_{max} = 27$ ms, $(\Delta_{offset})_{max} = 216$ ms, and **d** $(t_{jitter})_{max} = 27$ ms, $(\Delta_{offset})_{max} = 432$ ms and **e–h** corresponding results for time evolution of all devices' conductances involved in the experiment. A single data point on right panels corresponds to the conductance measured at the end of a particular epoch. For all experiments, $p_{spike} = 1$. All inputs and corresponding conductances are shown using unique colours. Thicker lines show synchronous inputs/weights

noise inputs, $(\Delta_{offset})_{max}$ was first set to 0 (Fig. 4a), which represents the simplest and more reproducible case with non-overlapping spikes. The $(\Delta_{offset})_{max}$ was then increased up to 432 ms (Fig. 4d), i.e., approximately 750% of the duration of pre-synaptic voltage spikes, in which case all 15 uncorrelated input spikes can fully overlap in time by chance and provide a current three times higher than the maximum current from the all-synchronous spikes.

Figure 5 shows coincidence detection results for another study, which demonstrate that the same neuron can learn a new pattern, while forgetting the previously learnt one. In this case, to simplify the experiment, $p_{spike}$ was set to 0.2, so that the average spiking activity was 0.238 Hz for each of the 15 noise inputs (Fig. 5a). $(\Delta_{offset})_{max}$ was set to 500 ms, so that, again, all of the noise spikes can overlap in time by chance. Note that the relative spiking activity of the noise inputs with respect to the synchronous ones was still much higher, and hence the considered task is more challenging, compared to those studied in ref. [28].

Figure 3 provides more details on the dynamics of the system by showing two representative voltage waveforms recorded by the oscilloscope. The recorded data in Fig. 3a shows that the pre-synaptic synchronous pulse and post-synaptic pulse overlapped in time. The former pulse was sent earlier in time, causing membrane potential to reach a firing threshold and, subsequently, generate the post-synaptic spike, which would be required to strengthen the synaptic efficacy. Obviously, for the considered task, this potentiation condition has happened more frequently and consistently for the synchronized inputs, so that their synaptic efficacies were reinforced. Figure 3b also shows an example of the synchronous pre-synaptic spike that was out of phase with other synchronous inputs. In this case, the membrane potential was below the firing threshold and leaked over time.

Finally, the additional results show that the coincidence detection experiment can be successfully performed multiple times using the same column (Fig. 6) and using different columns in the crossbar (Fig. 7).

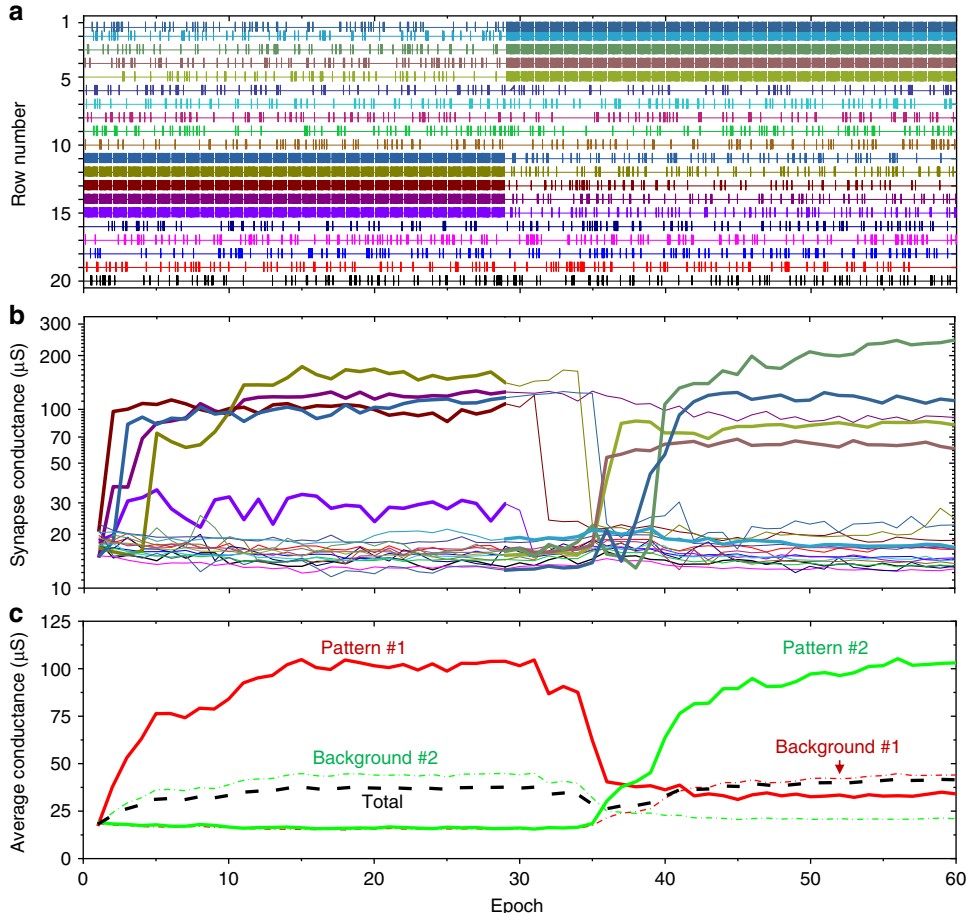

**Fig. 5** Coincidence detection of two lower-noise patterns. **a** A spiking input with $p_{spike} = 0.2$, $(t_{jitter})_{max} = 5$ ms, $(\Delta_{offset})_{max} = 500$ ms. The first pattern of synchronous spikes was applied to the rows #11 through #15 in the first 30 epochs, while the second pattern was applied to rows #1 through #5 in the last 30 epochs. **b** Time evolution of all devices' conductances involved in the experiment. **c** Post-processed results from panel **b**, showing time evolution of the total and the averaged conductances for all devices and the averages of subsets corresponding to the synchronized (pattern #1 and pattern #2) and noise inputs. The colour coding in panels **a** and **b** are similar to those of Fig. 4

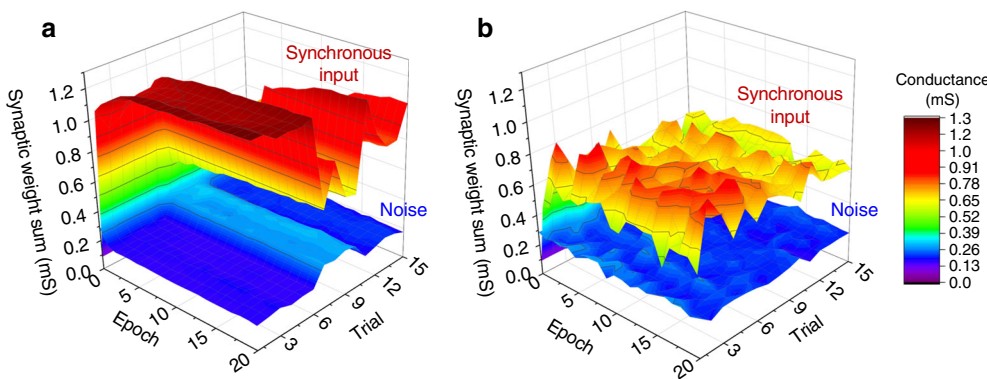

**Fig. 6** Trial statistics. Coincidence detection experiment repeated 15 times when connecting LIF neuron to **a** column #2 with $p_{spike} = 1$, with no jitter or offset added; and **b** column #19 with $p_{spike} = 1$, $(t_{jitter})_{max} = 25$ ms, and $(\Delta_{offset})_{max} = 200$ ms. The noise spikes were generated randomly and were unique for each frame in all trials

## Discussion

In both sets of experiments, the synaptic weights did not always evolve in the proper direction e.g., synchronized input #15 in Fig. 4. Additionally, the conductance change is not monotonic, with clearly noticeable jumps for some of synapses. This is partially due to random post-synaptic spikes generated by the output neuron in response to noise, but also due to non-idealities in memristors, most importantly their device-to-device variations in dynamic current−voltage characteristics. Nevertheless, the results in all experiments clearly show the progressive cumulative potentiation of the synapses that correspond to the synchronized inputs.

For example, Fig. 5c shows that when the second pattern was first applied (beginning of the epoch 31) the average synaptic strength corresponding to the synchronized inputs was initially

well below the average across all of the devices and more than two times lower than that of the uncorrelated inputs. However, as the training continued, the network was able to learn to discriminate those synchronized inputs by increasing their weights, eventually reaching and surpassing the average strength of uncorrelated inputs. In fact, by the end of the training, even the cumulative strength for the uncorrelated inputs was substantially lower compared to the synchronized ones for both patterns (Fig. 5c). In addition, as expected, the convergence took longer for the truly random noise input (Fig. 4h), because of the higher probability of firing a post-synaptic spike based on the spiking activity on the uncorrelated inputs (cf. with Fig. 4e).

Our experimental work has confirmed one of the main challenges for SNN implementation with memristors—their device-to-device variations[37,38]. Indeed, given similar pre- and post-synaptic spikes, the conductance updates should ideally follow the same STDP window for all synapses. In reality, the updates vary significantly across memristive synapses (Fig. 8), primarily due to variations in switching thresholds of the devices. Indeed, SNNs typically rely on fixed-magnitude spikes. In SNN's crossbar circuit implementation, the fixed-magnitude spikes are intended to update in parallel weights of multiple devices. In our experiment, the specific spike amplitudes were selected according to the average switching thresholds across the devices. Consequently, the change in the conductances for the devices with

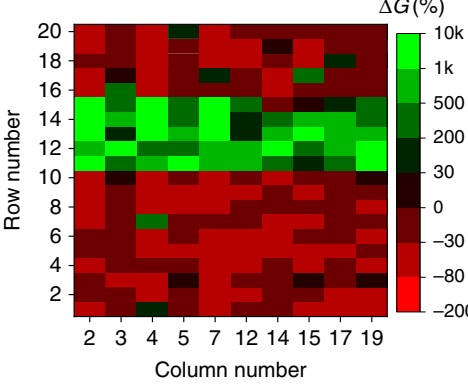

**Fig. 7** Column statistics. The relative change in device conductances after 20 epochs of applying synchronous input to rows #11 through #15, when connecting LIF neuron to a specified column in the crossbar. The experiments were performed separately for each column. $p_{spike} = 1$, $(t_{jitter})_{max} = 5$ ms, $(\Delta_{offset})_{max} = 27$ ms. The noise spikes were always generated randomly for each row in all frames

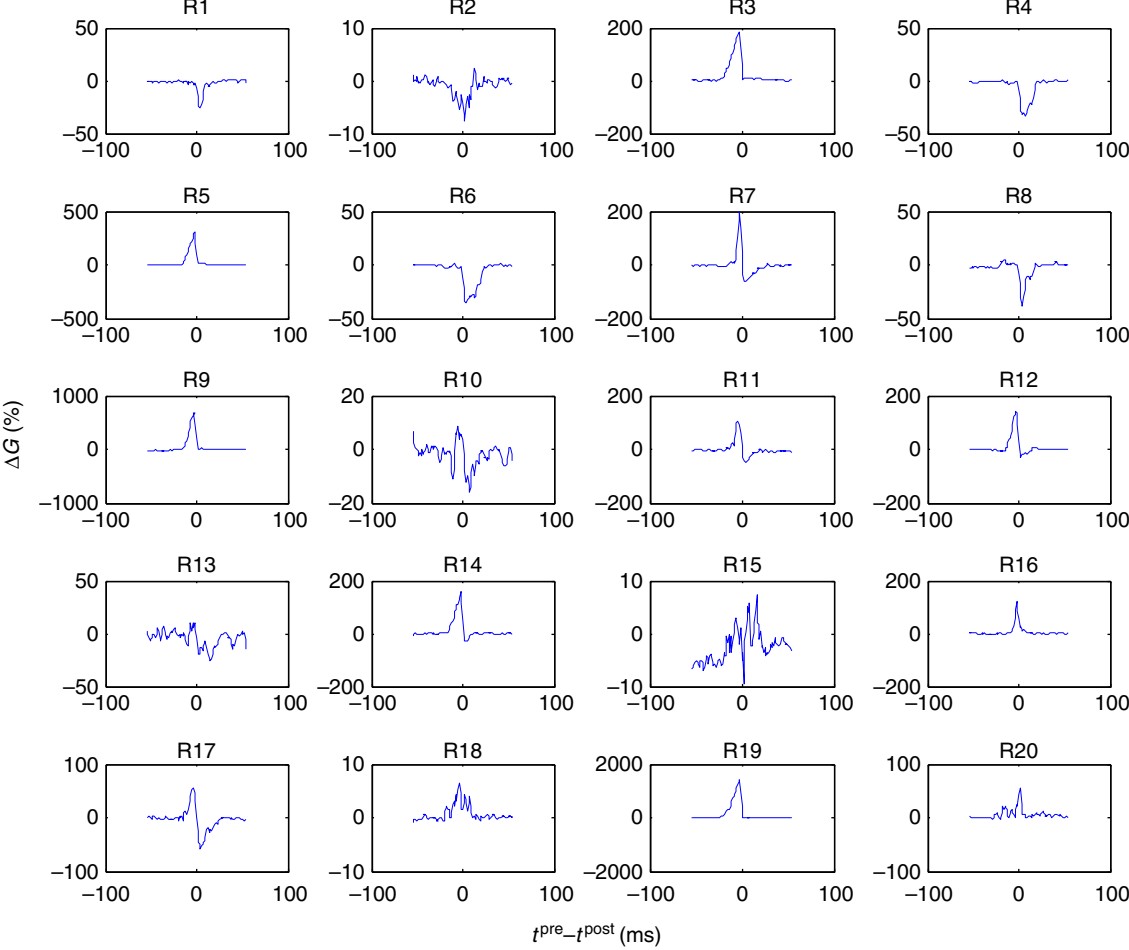

**Fig. 8** STDP variations. Measured STDP device-to-device variations for 20 memristors. The vertical axis shows the relative change in the conductance for a particular time difference between pre and post-synaptic spikes. (Note the different vertical axis scale for the panels.) The top label shows the device index, counted from the top of the crossbar circuit. Similar to ref. [12], the device conductance was always tuned to approximately 16 μS before measuring each data point

larger switching thresholds is naturally smaller, e.g., device #18 in Fig. 8, and alternatively, larger for those with smaller threshold (device #19). For the same reason, some STDP windows appeared to be noisy, e.g., that of device #15 in Fig. 8. In this respect, memristor implementation of the simpler ex situ trained firing-rate neural networks is much less challenging, as the write amplitude voltages in these networks can be adjusted uniquely for each device based on the feedback information during conductance tuning[9,38].

The STDP window for many memristors is also not balanced with either dominant potentiation (such as device #6 in Fig. 8) or depression (device #16). This is in part due to device-to-device variations in the on/off dynamic range—see, e.g., impact of the initial state on STDP function shape in ref. [12]. However, these variations only result in slight variation in the synaptic equilibrium currents, i.e., the memristor states reached after application of random spiking input[12], and hence, are not very critical for the network operation.

Recently, several groups reported promising devices with very uniform switching characteristics[39,40], that would naturally alleviate the device variation issue. An even better solution would be to holistically optimize training algorithms, circuits, and architectures with respect to device variations. For example, learning rules based on stochastic binary synapses[41,42] would be more robust to the non-ideal analogue switching. The device variations could also be addressed by considering more complex implementation of synapses[22,43] and/or having redundant devices in the network[44], though at the expense of larger area and power consumption.

In fact, our choice of the rectangular-shaped pulses utilized for pre- and post-synaptic voltage spikes is one example of the holistic optimization. Such shapes were selected in part for their simplicity of implementation but, more importantly, to ensure more reproducible STDP windows. In principle, the use of triangular pulses (e.g., those shown in Fig. 1a, b and implemented in ref. [12]) would cause smoother STDP curves, i.e., with more gradual changes in conductance. However, we found that such schemes resulted in less reproducible STDP behaviour when there are significant device variations. This is likely due to the specific switching dynamics of the considered devices (which is representative for many metal-oxide memristors[7,36]). Specifically, when the spikes are implemented with triangular pulses, the dropped voltage over the device also has a triangular waveform (see, e.g., Fig. 2a from ref. [12]), and hence maximum voltage is applied only over a short period of time. By contrast, with rectangular pulse implementation, the maximum voltage is applied over a longer time, which is comparable to pulse duration when pre- and post-synaptic spikes are fully overlapped in time. Because of quasi DC operation, i.e., relatively long pulse durations compared to the intrinsic switching times, and exponential switching dynamics with respect to the applied voltage, which saturates in time, the latter scheme seems to be more tolerant to memristor variations.

In conclusion, we have developed a complete SNN in which artificial synapses were implemented with passive integrated metal-oxide memristors, while LIF neuron was realized with discrete conventional semiconductor circuits. A specific but representative task for SNN, i.e., coincidence detection, which involves training a neuron to discriminate between synchronized inputs from uncorrelated ones, was successfully demonstrated by employing spike-timing-dependent plasticity learning. Our work showed that device-to-device variation in memristors' switching thresholds is the major challenge. Finally, we discussed several approaches for overcoming this challenge to build more practical, efficient memristor-based SNNs.

## Methods

**Spiking neural network implementation**. Artificial synapses are implemented with $Pt/Al_2O_3/TiO_{2-x}/Pt$ memristors, which were passively integrated in a $20 \times 20$ crossbar array (Fig. 2b). (A detailed discussion of the fabrication methods and electrical characterization results can be found in refs. [8,9].) The crossbar array was packaged (Supplementary Figure 2c) and inserted in a custom-printed circuit board (Supplementary Figure 2a) that allows connecting the crossbar lines either to the input/output neurons during network operation or to a switch matrix, which in turn is connected to a Keysight B1500A parameter analyser, for device forming, testing, and conductance tuning.

The input neurons, one for each row of the crossbar array, were implemented using off-the-shelf digital-to-analogue converter circuits. The output LIF neuron was connected to the third column of the crossbar array for the experiments reported in Figs. 3–5. (Other columns of the crossbar were grounded.) The neuron was implemented with a combination of another dedicated custom-printed circuit board (Supplementary Figure 2b), whose circuitry is shown in Fig. 2c, and an arbitrary waveform generator. The design of the neuron allowed fine-tuning its characteristic time, membrane potential threshold and the scale of synaptic current by adjusting the neuron's variable resistors. Operational amplifiers, TL074CN, were used in all three stages, while typical neuron circuit parameters are as following: $R_1 = 5.6$ KΩ, $R_2 = 2.5$ KΩ, $R_L = 900$ KΩ, $V_{b1} = 10$ mV, $V_{b2} = 0.8$ V and $C = 0.1$ μF.

Upon reaching a threshold, an LIF neuron triggers an arbitrary waveform generator. The latter was used to produce the post-synaptic spikes and, by using its 'sync' output, which is a transistor−transistor logic pulse synchronized with the signal output, to temporarily connect its signal output to the crossbar column and reset the neuron membrane potential. The entire experimental setup was controlled from a personal computer using a custom C program. The program controlled the generation of the pre-synaptic spikes by digital-to-analogue converters, an additional oscilloscope which was used to measure the inputs, post-synaptic spikes, and neuron membrane potential, and a switch matrix and parameter analyser.

**Spike pulse details**. For the experimental results shown in Figs. 3–7, the utilized pre-synaptic voltage spike consists of 5-ms ramp to +0.45 V, 7-ms 0.63 V plateau, 1-ms ramp down to –0.4 V, 20-ms −0.4 V plateau and then 5-ms ramp up to 0 V (Fig. 2). The post-synaptic voltage spike was 5-ms ramp from −0.15 V to +0.65 V, 23-ms +0.6 V plateau, 1-ms ramp down to –1.15 V, 22-ms −1.15 V plateau, and 5-ms ramp up to −0.15 V (Fig. 2a). The constant −0.15 V in the post-synaptic spike was for convenience of neuron implementation and not essential to the network operation. The total durations of pre-synaptic and post-synaptic pulses were 38 and 57 ms, respectively.

The results in Fig. 8 were obtained using a different $20 \times 20$ crossbar circuit, which was implemented using the same fabrication process. For this particular experiment, the pre-synaptic voltage spike shape was 5 ms ramp to +0.63 V, 10 ms to 0.63 V plateau, 2 ms ramp down to −0.54 V, 20 ms to −0.54 V plateau and then 5 ms ramp up to 0 V. The post-synaptic voltage spike shape was 5 ms ramp to +0.60 V, 23 ms to +0.6 V plateau, 2 ms ramp down to −0.5 V, 22 ms to −0.5 V plateau, and 5 ms ramp up to 0 V. The total durations of pre-synaptic and post-synaptic pulses were 42 and 57 ms, respectively.

## Data availability
The data that support the plots within this paper and other findings of this study are available from the corresponding author upon reasonable request.

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

## Acknowledgements
This work was supported by ARO under contract #W91NF-16-1-0302 and DARPA under contract HR0011-13-C-0051UPSIDE via BAE Systems, Inc.

## Author contributions
M.P., F.M.B., and D.S. conceived the original concept and initiated the work. M.P., H.N., and H.K. fabricated memristors. F.M.B., M.P., and A.V. developed the experimental setup. M.M., M.P., and H.K. performed measurements. D.S. and M.P. wrote the manuscript. All authors discussed the results.

## Additional information

**Competing interests:** The authors declare no competing interests.

