## [Peer Review File · Nature Communications]

Reviewers' comments:

Reviewer #1 (Remarks to the Author):

The paper demonstrates experimentally the connection of a single column of 20 memristive synapses distributed by columns (out of a 20x20 passive memristor array) to an analog integrate-and-fire neuron implemented in a pcb with discrete components.

The authors apply input pre-synaptic pulses by rows to each synaptic element and the signals of the output pulses are integrated in the column shared leaky integrate and fire neuron. By modulating the shape of pre- and post-synaptic pulses, they demonstrate the memristors undergo a spike-timing-dependent plasticity learning and measure the stdp learning characteristic curve for the 20 synaptic devices. Similar stdp curves have been already published by other groups for different memristive devices.

The authors use the experimental set-up to perform a simple coincidence detection task. They apply pulses sequentially to each synaptic element (pulses are separated 20 ms) but periodically they apply simultaneous synchronous spikes to 5 rows. The output neuron spikes when the synchronous pattern is applied. Authors state that 'coincidence detection, i.e. training a neuron to discriminate between synchronized input from asynchronous ones, was successfully demonstrated by employing spike-time-dependent-plasticity'. However, this is not really demonstrated by this simple task. Coincidence detection arises naturally in leaky integrate and fire neurons. Spikes that come spaced in time (as it is the case of the 20 ms spaced ones) tend to be forgotten by the neuron leakage. When several spikes arrives synchronously (tightly concentrated in time compared with the leakage time constant) they tend to produce an output spike. With the experiment in the paper, it is not shown or demonstrated that the synchrony detection comes as a consequence of learning.

To demonstrate that the network is able to learn and discriminate a particular synchronous pattern, authors should set-up at least two output neurons and demonstrate the stable learning and differentiation of 2 different input patterns by the two neurons.

Other questions to the authors:

-In figure 2, you show the voltage-time shapes applied as pre- and post-synaptic spikes to induce stdp. Why do pre- and post- synaptic spikes have so dissimilar shapes?. How do these pulse shapes have been optimized?

- coming back to the synchrony detection issue, what happens if noise events arrive near in time to several rows not corresponding to the learnt pattern, is the network able to distinguish them from the correct pattern?

- In supplementary figure 1, you show stdp curves variations from device to device. Do you have measurements of cycle to cycle variations for the same device? Or a similar question, what is the repeatability of these measurements?

Reviewer #2 (Remarks to the Author):

In the manuscript, the authors reported a STDP of coincidence detection with memristor arrays. The result seems to be interesting overall. However, the manuscript is not well organized. For example, introduction is two pages while result and discussion are one page each. The manuscript can be reconsidered for publication after major issues outlined below are adequately addressed.

1. The authors have only one page result and one page discussion without having detail explanations. Authors must restructure the manuscript completely. They need to describe in detail about the novelty of their data and their thought process. Also, the figure numbers must be addressed.
2. Can the authors explain why figure 2(technically, figure 3) is important? Can the authors clarify the message the authors want to deliver? It should be a part of previous figure 2(technically).
3. To strengthen this paper, can the authors show any function with this hardware?
4. For supplementary figure 2, the authors should explain the parts.
5. In conclusion part, "Our preliminary simulation results, based on accurate device models [44, 45], show that a reasonable increase in the number of synapses might be sufficient to deal with device to device variations for coincidence detection task." Please state this sentence again quantitatively. How many number of synapses are sufficient?

Reviewer #3 (Remarks to the Author):

The manuscript describes a significant, albeit modest step toward the implementation of practically valuable spiking neural networks based on memristive synapses. I believe that the paper may be published in Nature Communications, after the writing deficiencies listed below have been corrected.

Abstract:

Page 1, line 2. The statement "...outperformed so far..." is true only for the pattern classification tasks, though this is exactly the field where most progress was reached recently.

Page 1, line 8: The word "memories" should be replaced with the expression "memory devices".

Introduction:

Page 2, line 2: The expression "driven in hope" probably should read "driven by hope".

Page 2, lines 5-8: The statement "...SNN's weight updates rules are local..." is too general; actually the locality depends on the network's function, and the updates in for some firing-rate networks may be local as well.

Page 2, lines 20-21: The expression "we assume that the resting potential of the membrane is always zero" should be replaced with something like "the resting potential is taken for the reference", because actually there are no assumptions here.

Page 2, last line: the word "practical" should be replaced with "practically valuable".

Page 3, The paragraph starting from "Here, we..." should include the OT1R term for the used devices, because in the earlier text, other devices were discussed using this nomenclature.

Results:

Page 4, lines 1-2: Give a more explicit reference to the paper(s) describing the method of initial tuning

of all synapses (to 16 μ S).

Page 4, line 5: Remove the word "epochs" from this sentence, because its use here makes this term synonymous with the "patterns of spikes".

Page 4, lines 7-8: It is not clear to this reviewer why the sequential spike shifts by 20 ms (i.e. by an interval shorter than the used STDP window) is a good approximation for an spike uncorrelated pattern. This point should be explained.

Page 4, line 10 from the bottom: "Figure 4d" should read "Figure 4c".

Page 4, line 8 from the bottom: The expression "overlapped for synchronized input" is not sufficiently clear, making the meaning of the paragraph's balance rather obscure.

Discussion:

Page 5, line 13: Another word should be used here instead of "asymmetric", because this term describes a perfect STDP function of the type shown in Fig. 1c.

Figures:

In the caption to Fig. 2a, the expression "each point corresponds to the maximum region of the applied voltage spike" should be replaced with something like "each vertical dash shows the moment when the corresponding spike reaches its maximum."

In Fig. 2, I would remove the arrow from the line connecting panels (b) and (d), because it shows a two-way, rather than one-way connection.

In Fig. 2d, the meaning of legends "diode threshold" and "neuron threshold" is not clear. Usually, a "threshold" means a point at a curve, not an electronic circuit.

In the caption to Fig. 4, "post processed" should read "post-processed".

References:

Instead of Ref. 4 (and perhaps Refs. 1, 2, 5 and 6 as well), it would be better to cite the monograph by W. Gerstner and W. Kistler, "Spiking Neural Models", Cambridge U. Press, 2002.

General:

The manuscript needs a thorough proofreading by a native English speaker, with a special attention to the proper use of articles.

The authors would like to thank reviewers for reading our manuscript so carefully. In response, we have substantially revised the manuscript and extended Supplementary Information. To address reviewers' comments, we had reproduced the original experiments using additional crossbar circuit. This time we also collected plentiful statistical data and performed new important experiments. We believe that we addressed all of the reviewers' comments, which helped to substantially improve our paper.

This rest of this document consists of 2 parts:

- (a) a brief list of the most significant changes and the new material in the manuscript
- (b) detailed responses to all referee's comments, which also document all substantive changes made in the text, figures and figure captions of the paper.

(a) Significant changes

We have reproduced the original results using additional crossbar chip (fabricated using exactly the same recipe as the original crossbar circuit) and performed several new important studies. Specifically, we have

- repeated the original experiment (new Fig. 4a,b)
- performed new experiments with added jitter to the synchronous inputs and randomly generated noise inputs (new Fig. 4c-i)
- performed new experiment by demonstrating that the neuron can be trained to learn different patterns (new Fig. 5)
- added trial and column statistics for coincidence detection experiment (new Supplementary Figure 4 and 5, respectively)

(b) Point by point response to reviewers' comments

For convenience, the original comments/suggestions made by the referees are typeset in black, while our responses are provided in blue.

Reviewer #1:

The paper demonstrates experimentally the connection of a single column of 20 memristive synapses distributed by columns (out of a 20x20 passive memristor array) to an analog integrate-and-fire neuron implemented in a pcb with discrete components. The authors apply input pre-synaptic pulses by rows to each synaptic element and the signals of the output pulses are integrated in the column shared leaky integrate and fire neuron. By modulating the shape of pre- and post-synaptic pulses, they demonstrate the memristors undergo a spiketiming- dependent plasticity learning and measure the stdp learning characteristic curve for the 20 synaptic devices.

This is accurate description of our work.

Similar stdp curves have been already published by other groups for different memristive devices.

We agree and we explicitly stated it in our paper - see, e.g., “There have been many experimental demonstrations of various STDP rules [12-18] and more advanced synaptic functionalities based on memristors [19-21] – see also comprehensive reviews in Refs. 22, 23”. We hope that it was already clear that STDP results are not claimed as novelty. Nevertheless, to avoid further confusion, we have removed original Figure 2 completely. Note that the same results that were originally shown in Figure 2 were/are also presented in Supplementary Figure S1.

The authors use the experimental set-up to perform a simple coincidence detection task. They apply pulses sequentially to each synaptic element (pulses are separated 20 ms) but periodically they apply simultaneous synchronous spikes to 5 rows. The output neuron spikes when the synchronous pattern is applied.

This is again an accurate description of our work.

Authors state that 'coincidence detection, i.e. training a neuron to discriminate between synchronized input from asynchronous ones, was successfully demonstrated by employing spike-timedependent-plasticity'. However, this is not really demonstrated by this simple task.

We partially, though not fully, agree with this comment – please see our extended reply to the comment below

Coincidence detection arises naturally in leaky integrate and fire neurons. Spikes that come spaced in time (as it is the case of the 20 ms spaced ones) tend to be forgotten by the neuron leakage. When several spikes arrives synchronously (tightly concentrated in time compared with the leakage time constant) they tend to produce an output spike. With the experiment in the paper, it is not shown or demonstrated that the synchrony detection comes at a consequence of learning. To demonstrate that the network is able to learn and discriminate a particular synchronous pattern, authors should set-up at least two output neurons and demonstrate the stable learning and differentiation of 2 different input patterns by the two neurons.

We respectfully disagree with the Referee on this point. The goal of our paper is to demonstrate “coincidence detection”, which can be viewed as a very simple pattern matching, but the functionality that Referee is mentioning (in his/her last sentence) is, what we believe is called “pattern classification” (see, e.g., W. Maass, “Networks of spiking neurons: The third generation of neural network models”, *Neural Networks* 10 (9) pp. 1659-1671, 1997). Pattern classification builds upon coincidence detection and is naturally more challenging to demonstrate, e.g. because it requires inhibitory connections for the case of two output neurons. (Also, the Reviewer may find interesting paper by Y. Chen *et al.* “The role of coincidence-detector neurons in the reliability and precision of subthreshold signal detection in noise.” *PloS One* 8 (2), art. e56822, 2013, which highlights the importance of the pure coincidence detection neurons, in conjunction with LIF neurons, and paper which discusses the comparison between coincidence detection and LIF Neurons in J. Kanev *et al.* “Integrator or Coincidence detector: A novel measure based on

the discrete reverse correlation to determine a neuron's operational mode" *Neural computation* 28 (10) pp. 2091-2128, 2016.)

However, we agree that comprehensive demonstration of coincidence detection functionality also requires showing that the previously learnt coincidence pattern can be forgotten and the same neuron can learn a new one. This new important result is now added to the manuscript – see new Fig. 5 and its discussion.

We hope the Referee will find satisfactory new results, which now also include new experiments with added jitter to the synchronous inputs, new experiments with truly random noise spikes, extensive statistics, and repeatability demonstrations.

Other questions to the authors:

-In figure 2, you show the voltage-time shapes applied as pre- and postsynaptic spikes to induce stdp. Why do pre- and post- synaptic spikes have so dissimilar shapes?. How do these pulse shapes have been optimized?

This is a good point and we have added more clarification to the main text. Specifically, we added to the first paragraph in the Results section: "The specific amplitudes and durations of voltage pulses were chosen to ensure balanced STDP windows, with approximately 50% to 200% maximum changes in conductance, for a majority of the devices - see, e.g., typical STDP curves in Supplementary Figure 1. For example, the post-synaptic spike has a larger amplitude negative pulse, which was needed owing to larger absolute values of reset switching thresholds compared to those of set transition."

We also added new paragraph to the discussion section: "In fact, our choice of the rectangular-shaped pulses utilized for pre- and post-synaptic voltage spikes is one example of the holistic optimization. Such shapes were selected in part for their simplicity of implementation but, more importantly, to ensure more reproducible STDP windows. In principle, the use of triangular pulses (e.g., those shown in Figure 1a, b and implemented in Ref. 12) would cause smoother STDP curves, i.e., with more gradual changes in conductance. However, we found that such schemes resulted in less reproducible STDP behaviour when there are significant device variations. This is likely due to the specific switching dynamics of the considered devices (which is representative for many metal-oxide memristors [7, 36]). Specifically, when the spikes are implemented with triangular pulses, the dropped voltage over the device also has a triangular waveform (see, e.g., Fig. 2a from Ref. 12), and hence maximum voltage is applied only over a short period of time. By contrast, with rectangular pulse implementation, the maximum voltage is applied over a longer time, which is comparable to pulse duration when pre- and post-synaptic spikes are fully overlapped in time. Because of quasi DC operation, i.e., relatively long pulse durations compared to the intrinsic switching times, and exponential switching dynamics with respect to the applied voltage, which saturates in time, the latter scheme seems to be more tolerant to memristor variations."

- coming back to the synchrony detection issue, what happens if noise events arrive near in time to several rows not corresponding to the learnt pattern, is the network able to distinguish them from the correct pattern?

We have repeated experiments that was originally reported in Figure 4, this time adding jitter to the synchronous spikes and randomly shifting "noise" spikes – please see new extensive results reported in new Figure 4 and its discussion. We believe that the case Referee is mentioning

frequently happens in both considered experiments reported in new Figures 4 and 5 – see e.g. noise spikes at rows 18 and 20 in Figure 4c. Such events naturally slow down the training. We have added corresponding short discussion in main text.

- In supplementary figure 1, you show stdp curves variations from device to device. Do you have measurements of cycle to cycle variations for the same device? Or a similar question, what is the repeatability of these measurements?

We have added new extensive results on repeatability and reproducibility of the experiment – please see new Figures 4 and 5, as well as Supplementary Figures 4 and 5.

Reviewer #2:

In the manuscript, the authors reported a STDP of coincidence detection with memristor arrays. The result seems to be interesting overall.

We would like to thank Referee for this comment.

However, the manuscript is not well organized. For example, introduction is two pages while result and discussion are one page each. The manuscript can be reconsidered for publication after major issues outlined below are adequately addressed.

We completely agree with this criticism and provided our responses for each point raised by the Referee below.

1. The authors have only one page result and one page discussion without having detail explanations. Authors must restructure the manuscript completely. They need to describe in detail about the novelty of their data and their thought process. Also, the figure numbers must be addressed.

To address this comment we have substantially extended Result and Discussion sections. In the revised manuscript, the combined length for Results/Discussion/Methods (which can be considered as a part of results) sections is now ~4.5 pages, and ~6 pages including figures, which we believe is a good balance.

2. Can the authors explain why figure 2 (technically, figure 3) is important? Can the authors clarify the message the authors want to deliver? It should be a part of previous figure 2 (technically).

Thank you for pointing to this typo. The original Figure 3 is completely removed. It has been replaced with updated and extended Figure 4c from the original manuscript. To improve quality of the manuscript, we have also updated some panels in Figure 2.

3. To strengthen this paper, can the authors show any function with this hardware?

We are sorry but we are not sure if we understand this comment. In this paper we demonstrate a “coincidence detection”. It is a simple task and by itself may not be practical. However, as we

discussed in the introduction, this simple task is believed to be at the core of many other, truly cognitive and complex functionalities.

4. For supplementary figure 2, the authors should explain the parts.

To address this comment, we have extended caption for Supplementary Figure 2.

5. In conclusion part, “Our preliminary simulation results, based on accurate device models [44, 45], show that a reasonable increase in the number of synapses might be sufficient to deal with device to device variations for coincidence detection task.” Please state this sentence again quantitatively. How many number of synapses are sufficient?

Since we cannot provide definitive quantitative results at this point (our simulation study is still ongoing) we have removed this sentence and references [45] and [46] from the original manuscript.

Reviewer #3:

The manuscript describes a significant, albeit modest step toward the implementation of practically valuable spiking neural networks based on memristive synapses. I believe that the paper may be published in Nature Communications, after ...

We would like to thank Reviewer for his/her assessment of our work. Please note that we have extensively updated SI and added new results to further strengthen the significance of our work.

... the writing deficiencies listed below have been corrected.

Abstract:

Page 1, line 2. The statement “...outperformed so far...” is true only for the pattern classification tasks, though this is exactly the field where most progress was reached recently.

We slightly rephrased the sentence in question as following: “Though outperformed so far in most practical applications by simpler...”

Page 1, line 8: The word "memories" should be replaced with the expression “memory devices”.

Corrected

Introduction:

Page 2, line 2: The expression “driven in hope” probably should read “driven by hope”.

Corrected

Page 2, lines 5-8: The statement “...SNN’s weight updates rules are local...” is too general; actually the locality depends on the network's function, and the updates in for some firing-rate networks may be local as well.

We modified text to address this comment as following: “Moreover, training for the firing-rate networks typically relies on a backpropagation algorithm, the efficient implementation of which is challenging owing in part to the centralized method for computing weight updates and the requirement for large high-precision memory. By contrast, most popular SNNs’ weight updates”

Page 2, lines 20-21: The expression “we assume that the resting potential of the membrane is always zero” should be replaced with something like “the resting potential is taken for the reference”, because actually there are no assumptions here.

We have modified the text to: “Additionally, Eq. (1) does not explicitly include the resting potential. Instead, the resting potential is used as a reference for $U(t)$.”

Page 2, last line: the word "practical" should be replaced with “practically valuable”.

Corrected

Page 3, The paragraph starting from “Here, we...” should include the OT1R term for the used devices, because in the earlier text, other devices were discussed using this nomenclature.

Corrected

Results:

Page 4, lines 1-2: Give a more explicit reference to the paper(s) describing the method of initial tuning of all synapses (to 16 uS).

We added new reference (36) and changed text to: “...were set to approximately 16 μ S (Fig. 4a) using automatic tuning algorithm [36]...”

Page 4, line 5: Remove the word “epochs” from this sentence, because its use here makes this term synonymous with the “patterns of spikes”.

Corrected. For clarity, we have also added more details to Fig. 2a, and added new Supplementary Figure 3.

Page 4, lines 7-8: It is not clear to this reviewer why the sequential spike shifts by 20 ms (i.e. by an interval shorter than the used STDP window) is a good approximation for an spike uncorrelated pattern. This point should be explained.

We agree with the Reviewer’s concern. It would have been more appropriate to call sequential spike shifts “a pattern without coincidental spikes”. To address this comment, we have added new extensive experimental results for coincidence detection, considering both original spiking input with sequential spike shifts, which results in more repeatable behavior of the system – see new Figure 4a, as well as spiking input with truly randomly generated “noise” spikes – see, e.g. Figure 4b-d, and also Supplementary Figure 3 explaining how we generate spike input in the new set of experiments.

Page 4, line 10 from the bottom: “Figure 4d” should read “Figure 4c”.

We have updated the original Figure 4 and made corresponding changes in text.

Page 4, line 8 from the bottom: The expression “overlapped for synchronized input” is not sufficiently clear, making the meaning of the paragraph’s balance rather obscure.

We have updated this Figure (which is now Figure 3) and substantially modified its discussion – see the last paragraph on page 4.

Discussion:

Page 5, line 13: Another word should be used here instead of “asymmetric”, because this term describes a perfect STDP function of the type shown in Fig. 1c.

We have changed text to: “STDP window for many memristors is also not balanced with either dominant potentiation (such as device #6 in Supplementary Figure 1) or depression (device #16).”

Figures:

In the caption to Fig. 2a, the expression “each point corresponds to the maximum region of the applied voltage spike” should be replaced with something like “each vertical dash shows the moment when the corresponding spike reaches its maximum.”

We believe that for updated Figure 2 this sentence is no longer needed, and we have removed it from the caption.

In Fig. 2, I would remove the arrow from the line connecting panels (b) and (d), because it shows a two-way, rather than one-way connection.

Corrected. We have also updated Figure 2 to improve its quality.

In Fig. 2d, the meaning of legends “diode threshold” and “neuron threshold” is not clear. Usually, a “threshold” means a point at a curve, not an electronic circuit.

We have removed these misleading labels.

In the caption to Fig. 4, “post processed” should read “post-processed”.

Corrected.

References:

Instead of Ref. 4 (and perhaps Refs. 1, 2, 5 and 6 as well), it would be better to cite the monograph by W. Gerstner and W. Kistler, “Spiking Neural Models”, Cambridge U. Press, 2002.

We have replaced Ref. 4 with the suggested one. To further address this comment original Ref. 4 is now Ref. 1 in revised manuscript and all reference numbers are updated accordingly.

General:

The manuscript needs a thorough proofreading by a native English speaker, with a special attention to the proper use of articles.

To improve writing quality, our revised manuscript has been proofread by Nature Research Editing Services.

Thank you very much again for reading our manuscript so carefully! All of these comments are now addressed in the revised version.

Reviewers' comments:

Reviewer #1 (Remarks to the Author):

The paper has been improved adding repetitive experiments and illustrating the effect of noise in the coincidence detection task. However, there are some issues that should still be addressed to improve the paper and make it deserve publication in a prestigious journal like this one.

In the abstract, authors still state that "by employing STDP learning, the implemented network is able to successfully detect the coincidence in the input signals coming from 5 of the 20 afferents", which is not true. The postsynaptic neuron is able to detect the coincidence of the 5 input signals before the STDP learning goes on. That demonstrates that the coincidence detection task arises naturally from the Integrate and Fire post synaptic neuron and it is not enabled by learning. The effect of STDP learning is just memorizing the pattern, which it just makes sense if this acquired learning is going to be used afterwards in a pattern classification task.

It is true that for a pattern classification task, inhibitory connections between postsynaptic neurons are needed to avoid all the neurons learn the same pattern. It is also true that it is a more challenging task. However, I do not understand what prevents the authors from implementing and demonstrating this task using at least two postsynaptic neuron PCBs with inhibitory connections among them. I think that, precisely because it is a more challenging task, such an implementation would put a real value on the paper.

I do not see that showing that the previously learnt pattern can be forgotten and the neuron can learn a new pattern demonstrates that the coincidence detection task is enabled by STDP learning. I would say it is the opposite. The neuron is still able to detect coincidence even for a different pattern which is not the one that it learned.

Reviewer #2 (Remarks to the Author):

Authors addressed issue appropriately. I do not have any further comments.

Reviewer #3 (Remarks to the Author):

I believe that at the revision the authors have fully addressed the issues mentioned in my first review. In addition, they (in response to other reviewers' comments) have included additional experimental data, which increase the work's value. As a result, I can endorse the publication of the manuscript in its current form.

For convenience, the original comments/suggestions made by the first Referee are typeset in blue, while our responses are provided in black.

The paper has been improved adding repetitive experiments and illustrating the effect of noise in the coincidence detection task.

Thank you for this comment. Please note that these additional experiments took very substantial effort.

However, there are some issues that should still be addressed to improve the paper and make it deserve publication in a prestigious journal like this one. In the abstract, authors still state that "by employing STDP learning, the implemented network is able to successfully detect the coincidence in the input signals coming from 5 of the 20 afferents", which is not true. The postsynaptic neuron is able to detect the coincidence of the 5 input signals before the STDP learning goes on. That demonstrates that the coincidence detection task arises naturally from the Integrate and Fire post synaptic neuron and it is not enabled by learning. The effect of STDP learning is just memorizing the pattern, which it just makes sense if this acquired learning is going to be used afterwards in a pattern classification task.

We strongly disagree again with the criticism that our results do not demonstrate STDP learning of coincidence detection. Clearly, the postsynaptic neuron must detect (e.g. based on LIF functionality only) some coincidence to turn on STDP learning mechanism. This would be also true for more complicated system, e.g. a system with inhibitory synapses performing pattern recognition that Referee has mentioned in his/her comments. The important point is that the coincidence detection is "weak" at first - the output neuron does not always fire when it receives correlated inputs, and its response is unstable in the presence of noise and/or jitter, e.g. resulting in firing many spikes at random. On the other hand, due to STDP learning mechanism, the neuron is gradually trained to respond more regularly and more robustly to synchronous inputs, especially in the presence of noise and jitter, by selectively strengthening synaptic conductances. Our main results (Figs. 4 and 5) clearly demonstrate this point.

We agree though that there might be some ambiguity on what to call "coincidence detection" functionality. In this regard, we believe that described above "weak" coincidence detection is not useful without further STDP learning. Specifically, in a more practical system, the output neuron would be a part of a larger network. For example, the output neuron would feed some other "super" neuron, which also has to perform coincidence but at the higher level of abstraction, e.g. correlating activities from different coincidence tasks. Having weak coincidence input would not be useful for the super neuron, just like very noisy and jittery inputs were problematic for our task (cf. Fig. 4e and 4i).

Interestingly, Referee also wanted to see "stable" learning in his/her comments to the original paper. See e.g. "To demonstrate that the network is able to learn and discriminate a particular synchronous pattern, authors should set-up at least two output neurons and demonstrate the stable learning and differentiation of 2 different input patterns by the two neurons." The stable learning is exactly what is achieved by STDP mechanism and what we demonstrate in our paper.

To make this point clear we have further slightly reworded abstract as:

Before: **By employing spike-timing-dependent plasticity learning, the implemented network is able to successfully detect the coincidence in the input signals coming from 5 of the 20 afferents and selectively increase the synaptic efficacies corresponding to the synchronized inputs.**

Now: **By employing spike-timing-dependent plasticity learning, the implemented network is able to robustly detect the coincidence in the input signals coming from 5 of the 20 afferents by selectively increasing the synaptic efficacies corresponding to the synchronized inputs.**

and added more discussion in the main text, as following:

Before: **Additionally, the conductance change is not monotonic, with clearly noticeable jumps for some of synapses, which is likely due to non-idealities in memristors, most importantly their device-to-device variations in dynamic current-voltage characteristics.**

Now: **Additionally, the conductance change is not monotonic, with clearly noticeable jumps for some of synapses. This is partially due to random post-synaptic spikes generated by the output neuron in response to noise, but also due to non-idealities in memristors, most importantly their device-to-device variations in dynamic current-voltage characteristics.**

It is true that for a pattern classification task, inhibitory connections between postsynaptic neurons are needed to avoid all the neurons learn the same pattern. It is also true that it is a more challenging task. However, I do not understand what prevents the authors from implementing and demonstrating this task using at least two postsynaptic neuron PCBs with inhibitory connections among them. I think that, precisely because it is a more challenging task, such an implementation would put a real value on the paper.

We appreciate the suggestion but we feel that this by itself would be a new large-scale year-long project. We invite Referee to look at Figs. 2 and S2 again to appreciate how complicated the system already is with one neuron. We would like to stress again that pattern recognition is different functionality and would not be required to demonstrate “stable” or “strong” coincidence detection. The focus of our work was/is on the coincidence detection, which by itself is very important functionality - see, e.g., discussion in the last two sentences in the introduction section.

We would also like to note that coincidence detection results for the most complex prior work that we are aware of (Ref. [28] in the revised manuscript) were achieved by post-processing experimental results from single devices based on inferior (less dense) 1T1R technology. Furthermore, that work involved digital integration and emulation of the neuron functionality in a software, and the applied noise was much less dense. On the other hand, in our work we show coincidence without any post-processing using the most prospective crossbar-integrated passive memristors under more challenging noisy background. Moving from single devices to an integrated system based on emerging devices, and its further linear increase in complexity, often implies exponential improvements in device technology, e.g. improvements in device yield.

I do not see that showing that the previously learnt pattern can be forgotten and the neuron can learn a new pattern demonstrates that the coincidence detection task is enabled by STDP learning. I would say it is the opposite. The neuron is still able to detect coincidence even for a different pattern which is not the one that it learned.

We again disagree since the Referee seems to imply “weak” coincidence detection in his/her last sentence. The importance of learning and forgetting experiment is to show that, in the demonstrated system, synaptic connections can be both significantly strengthened and weakened via STDP learning, which would be certainly essential for more sophisticated memristor-based SNNs.

REVIEWERS' COMMENTS:

Reviewer #1 (Remarks to the Author):

The authors have done very minor revisions on the previous version.

The authors state that with their experiments they demonstrate that the network successfully increases the synaptic efficacies corresponding to the synchronized input making the network more robustly respond to that pattern what it is true.

They also argue that their work is based on the use of a denser crossbar of devices than previous works. However, this reviewer considers that the contribution is still marginal because they do not exploit and do not demonstrate the functionality of such a crossbar as their work is based on the use of a single column. Furthermore, this reviewer considers that the addition of a second PCB hosting a second post-synaptic neuron adds a reasonable increment of complexity to the system.

Consequently, this reviewer is not satisfied with the author's answers and cannot recommend the paper for publication.

Of course, as the other two reviewers are satisfied with the work, it is under the criterion of the associate editor to consider the work acceptable for publication.

Reviewer #2 (Remarks to the Author):

As previously informed, the manuscript is well addressed and ready to be published. The manuscript could have been much more improved if the authors could complete the task requested by the reviewer 1. In my point of view, the paper is meaningful enough in the current form as authors show feasibility of the STDP learning of coincidence detection with memristors. Demonstration beyond current level could be considered for a next separate paper.

Reviewer #3 (Remarks to the Author):

I agree with the authors' reply to Reviewer #1. Indeed, this work clearly shows that the used STDP learning mechanism makes the neuron to respond more regularly / robustly to learned input patterns. (Fig. 5 illustrates this point very clearly.) The last changes of the manuscript's text have made this point more clear. I would favor the work's publication in the current form.